# Supplementation with a Highly Concentrated Docosahexaenoic Acid (DHA) in Non-Proliferative Diabetic Retinopathy: A 2-Year Randomized Double-Blind Placebo-Controlled Study

**DOI:** 10.3390/antiox11010116

**Published:** 2022-01-05

**Authors:** Purificación Piñas García, Francisco Javier Hernández Martínez, Núria Aznárez López, Luis Castillón Torre, Mª Eugenia Tena Sempere

**Affiliations:** Service of Ophthalmology, Hospital San Juan de Dios del Aljarafe, Avenida San Juan de Dios s/n, E-41930 Bormujos, Sevilla, Spain; fjhernandezm@hotmail.com (F.J.H.M.); nuazlo@yahoo.es (N.A.L.); lcastillon@telefonica.net (L.C.T.); draojitos@yahoo.es (M.E.T.S.)

**Keywords:** non-proliferative diabetic retinopathy, docosahexaenoic acid, antioxidant, randomized controlled study

## Abstract

We assessed the effect of a 2-year supplementation with a highly concentrated docosahexaenoic acid (DHA) product with antioxidant activity on non-proliferative diabetic retinopathy (NPDR) in a randomized double-blind placebo-controlled study. A total of 170 patients with diabetes were randomly assigned to the DHA group (*n* = 83) or the placebo group (*n* = 87). NPDR was diagnosed using non-contact slit lamp biomicroscopy examination, and classified into mild, moderate, and severe stages. Patients in the DHA group received a high rich DHA triglyceride (1050 mg/day) nutritional supplement, and those in the placebo group received olive oil capsules. The percentages of mild NPDR increased from 61.7% at baseline to 75.7% at the end of the study in the DHA group, and from 61.9% to 73.1% in the placebo group. Moderate NPDR stages decreased from 35.1% at baseline to 18.7% at the end of the study in the DHA group, and from 36.8% to 26.0% in the placebo group. In the DHA group, there were five eyes with severe NPDR at baseline, which increased to one more at the end of the study. In the placebo group, of two eyes with severe NPDR at baseline, one eye remained at the end of the study. Changes in visual acuity were not found. There were improvements in the serum levels of HbA1c in both groups, but significant differences between the DHA and the placebo groups were not found. In this study, the use of a DHA triglyceride nutraceutical supplement for 2 years did not appear to influence the slowing of the progression of NPDR.

## 1. Introduction

Non-proliferative retinopathy (NPDR) is the earliest stage of diabetic retinopathy (DR), in which symptoms can be mild or non-existent. NPDR typically involves microvascular changes and progresses from mild to moderate and severe stages, and, in some people, may progress to sight-threatening DR, such as proliferative diabetic retinopathy (PDR) and diabetic macular edema (DME). Hyperglycemia, hypertension, and increasing duration of diabetes are independent risk factors for DR, and although improved glycemic control and blood pressure, regularly monitoring of DR, and intensive treatment of modifiable risk factors are crucial to prevent complications [1,2,3], clinically this is difficult to achieve [4].

About one-third of the diabetes population suffers from DR, mostly NPDR, but there are scanty data on the prevalence of NPDR. In a review of 32 studies with 543,448 people with diabetes who underwent retinal photography as a basis for diagnosing the presence and severity of DR, the overall prevalence of NPDR was 19% (range 11.7–65%) [5]. In a follow-up study of patients with mild NPDR, the cumulative occurrence rate of PDR at 10 years was estimated to be 14% in subjects with a mean HbA1c <8.6%, increasing to 60% in those with HbA1c ≥8.6% [6]. Systematic reviews and meta-analyses have shown a pooled prevalence of NPDR of 15.1% in patients with diabetes and 27% in patients with type 2 diabetes [7,8]. In a recent comprehensive review based on 90 studies with 204,189 patients with diabetes, a prevalence of NPDR of 24.8% was reported [9]. Approximately half of the patients with severe NPDR will progress to PDR within a year [4], so the prevention of disease progression to more advanced stages with potential visual loss and the identification of people at high risk of progression and greatest potential to benefit from treatment is of the utmost importance.

The treatment of mild NPDR may not be necessary, but understanding the multifactorial pathogenic mechanisms of DR could optimize the treatment of moderate/severe NPDR and slow the progression to PDR or DME [10]. Several mechanisms, by which hyperglycemia causes retinal capillary damage, include increased polyol and hexosamine pathways, increased non-enzymatic glycation with advanced glycation end-products (AGE), abnormal activation of signaling cascades such as protein kinase C pathway (PKC), increased oxidative stress, increased expression of adhesion molecules, local inflammatory activity with upregulation of proinflammatory mediators, interleukins, and critically important growth factors promoting angiogenesis, breakdown of the blood–retinal barrier, and retinal neurodegeneration [11,12,13,14,15,16]. Omega-3 long-chain polyunsaturated fatty acids (*n*-3 PUFAs), particularly docosahexaenoic (DHA) acid, have demonstrated consistent anti-inflammatory, antiproliferative, antiangiogenic, and antioxidant properties on pathways leading to DR [17]. These include promoting vascular integrity [18], reduction of oxidative stress-induced apoptosis of photoreceptors [19], neuroprotection [20], inhibition of the nuclear factor-kappa B (NF-κB) signaling pathway [21], reduction of proinflammatory cytokine production (IL-6, IL-10, IL-1β, FNTα) and intercellular and vascular adhesion molecules [22], reduction of pathological retinal angiogenesis [23], and increase of E3-derived anti-inflammatory mediators (E3 eicosanoids, B5 leukotrienes) and pro-resolving mediators (protectin D1, resolvins E1, D1) [24].

These pleomorphic effects of DHA support the rationale of dietary supplementation with DHA in DR. However, there is little experience in the use of dietary supplementation with DHA in early stages of DR. In a prospective controlled study of 12 asymptomatic patients with NPDR and 12 healthy controls, high rich DHA triglyceride (DHA-TG) (1050 mg/day) supplementation for 90 days was associated with the progressive and significant improvement of macular function measured by microperimetry in eyes from DHA-treated subjects compared with controls [25].

To our knowledge, clinical studies on the effect of DHA dietary supplementation with antioxidant activity in diabetic patients with NPDR have not been previously reported. Therefore, a randomized controlled study was conducted to determine whether dietary supplementation with high dose DHA for 2 years could slow the progression of any pre-proliferative stage of NPDR in patients with diabetes. 

## 2. Materials and Methods

### 2.1. Study Design and Participants

A prospective randomized double-blind placebo-controlled study (PAOXRED study, “Protección AntiOXidante en la REtinopatía Diabética”, Antioxidant Protection in Diabetic Retinopathy) was conducted at the Service of Ophthalmology of a regional hospital in Sevilla, Spain. Type 1 and type 2 diabetic patients of both sexes aged >18 years, diagnosed with NPDR (any stage) by four specialized ophthalmologists (P.P.G., F.J.H.M., N.A.L., E.T.S.), were invited to participate in the study during a routine ophthalmological appointment at the study center. NPDR was diagnosed using non-contact slit lamp biomicroscopy examination, and classified into mild, moderate, and severe stages, in the absence of neovascularization [26]. Mild NPDR was characterized by microaneurysm(s) only; moderate NPDR by at least one hemorrhage or microaneurysm and/or at least one of the following: retinal hemorrhages, hard exudates, cotton-wood spots, venous beading; and severe NPDR by any of the following but no signs of PDR (4-2-1 rule): >20 intraretinal hemorrhages in each of the four quadrants, definite venous beading in two or more quadrants, and prominent intraretinal microvascular abnormality (IRMA) in one or more quadrants. The exclusion criteria were as follows: the presence of PDR and/or DME documented on optical coherence tomography (OCT), previous surgery for morbid obesity, chronic diarrhea of any cause, anticoagulation, known allergy to fish proteins, use of dietary supplementation with vitamin/minerals or fatty acids, pregnant women, cognitive impairment, patients unable to participate according to the criteria of the ophthalmologist, and those who refused to give written consent.

This study protocol was approved by the Clinical Research Ethics Committee of Hospital San Juan de Dios del Aljarafe (Sevilla, Spain) (study code PAOXRED Vo2, approval date 20 January 2017). All participants provided written informed consent.

### 2.2. Study Intervention

Eligible patients were assigned to the DHA supplementation group (experimental) or the control group using pseudorandom numbers generated by the data collection computer server at the time of entering the first datum of a patient, with *p* = 0.5 so that each patient had a 50% probability of being randomized to one of the two study groups. Randomization was implemented without restriction or additional procedures to balance the sample size in each study group.

The patients in the DHA group received a high rich DHA triglyceride (1050 mg/day) nutraceutical formulation (Brudyretina 1.5 g, Brudy Lab, S.L., Barcelona, Spain). This is a concentrated DHA triglyceride having a high antioxidant activity patented to prevent cellular oxidative damage [27]. Table 1 shows the composition of the nutraceutical formulation, which includes DHA, eicosapentaenoic acid (EPA), vitamins (B-complex, C, E), lutein, zeaxanthin, glutathione, and minerals. 

Patients in the placebo group were treated with identically appearing olive oil capsules (also labeled Brudyretina 1.5 g). All patients were instructed to take three capsules of Brudyretina 1.5 g once daily, preferably in the morning with food and a glass of water.

### 2.3. Study Procedures

Patients were recruited between March 2017 and December 2020, and were followed for 24 months, with control visits every 6 months. At the baseline visit, the inclusion criteria were checked, the written informed consent was signed, and the nutraceutical formulation (Brudyretina 1.5 g) was delivered for the initial 6-month treatment period. 

The ophthalmological studies included non-contact (Volk SuperField NC^®^) slit lamp biomicroscopy examination of the optic fundus with mydriasis, measurement of best-corrected visual acuity (BCVA) using an ETDRS optotype at 2 m distance from the observer, and assessment of the macular condition by optical coherence tomography (OCT) (Stratus OCT, Carl Zeiss Meditec, Dublin, CA, USA). Retinography was also performed, as well as OCT to exclude DME. Visual acuity (VA) was expressed in a decimal scale and in logarithm of the minimum angle of resolution (logMAR). A peripheral venous blood sample was drawn after at least 8 h fasting to measure the serum levels of glycosylated hemoglobin (HbA1c) as an indicator of metabolic control, which was defined as an HbA1c value of 7–8% following the clinical practice guidelines of redGDPS (Network of Diabetes Study Groups in Primary Care) for patients with long-standing diabetes or comorbidity [28]. The serum levels of HbA1c were measured using a high-performance liquid chromatography (HPLQ) analyzer (Menarini Diagnostics, Badalona, Barcelona, Spain).

The same examinations were performed at each study visit, at 6, 12, 18, and 24 months after enrolment. At the 6-, 12-, and 18-month visits, the nutraceutical formulation was delivered for the next 6-month treatment period. The ophthalmologists paid special attention to insist on the importance of compliance with the dietary supplement and the benefit that the patient may receive from the supplement. At each visit, the patients were interviewed about gastrointestinal tolerability to the nutraceutical formulation and other side effects. Compliance with the nutraceutical supplementation was assessed at the study visits by return of supplementation tablet counts. The ophthalmologists who evaluated the results of the treatment (P.P.G., F.J.H.M., N.A.L., and E.T.S.) were blinded to which subjects were assigned to the experimental or the placebo group. All data were anonymized and recorded by the researchers on a specific website with access through a personal password.

### 2.4. Outcomes

The primary outcome was the change of the stages of NPDR during the study in the DHA and the placebo groups, in particular the number of eyes in each NPDR stage at 24 months compared with baseline. Changes of the stages of NPDR were also assessed in terms of improvement (decrease in any NPDR stage), unchanged (remaining in the same NPDR stage), and worsening (increase in any NPDR stage). Changes of VA and the serum levels of HbA1c were the secondary outcome variables.

### 2.5. Statistical Analysis

The per-protocol (PP) data set was analyzed—that is, all randomized patients who attended the follow-up visits and complete the 2-year study period. Categorical variables are expressed as frequencies and percentages, and continuous variables as mean and standard deviation (SD). Categorical variables were compared with the chi-square (χ^2^) test or the Fisher’s exact test according to the conditions of application. Quantitative variables were compared with the non-parametric Mann-Whitney *U* test. Within-group differences of VA and the serum levels of HbA1c during the study period were analyzed with the Wilcoxon signed-rank test, and changes of NPDR stages using the McNemar–Browker test. Statistical significance was set at *p* < 0.05. Statistical analyses were performed with the Statistical Package for the Social Sciences, version 26.0 software (IBM Corp., Armonk, NY, USA).

## 3. Results

### 3.1. Characteristics of the Study Patients

A total of 170 patients with diabetes (type 1, 15 patients; type 2, 155 patients) were included in the study and were assigned to the DHA group (*n* = 83) or the placebo group (*n* = 87). There were 130 men and 40 women, with a mean age (SD) of 61.7 (11.3) years. As shown in Table 2, there were no significant differences in the baseline characteristics of the patients assigned to the DHA or the placebo groups. More than 60% of the patients from each group had mild NPRD, and only 3.2% and 1.3% of patients in the DHA and placebo groups, respectively, presented with a severe stage. Visual acuity and the serum levels of HbA1c were similar in both study groups. Most patients received antidiabetic treatment, with oral antidiabetic drugs administered in half of the patients and combined with insulin in 30% of cases.

A total of 83 patients (154 eyes) were assigned to the DHA group and 59 (71.1%) patients (107 eyes, 69.5%) completed the study, whereas of the 87 patients (163 eyes) assigned to the placebo group, 63 (72.4%) (119 eyes, 73.0%) completed the study (Figure 1). However, statistically significant differences in the baseline characteristics between the patients who were lost to follow-up and those who completed the study were not found (Table 3).

### 3.2. Changes of NPDR Stage

In relation to the primary outcome of the study, the percentages of mild NPDR increased from 61.7% at baseline to 75.7% at the end of the study in the DHA group, and from 61.9% to 73.1% in the placebo group. Moderate NPDR stages decreased from 35.1% at baseline to 18.7% at the end of the study in the DHA group, and from 36.8% to 26.0% in the placebo group. In the DHA group, there were 5 eyes with severe NPDR at baseline, which increased to 1 more at the end of the study. In the placebo group, of 2 eyes with severe NPDR at baseline, 1 eye remained at the end of the study. As shown in Table 4, differences of changes in NPDR stages at each study visit as compared with baseline were not statistically significant in any of the study groups, except for the within-group comparison of visits at 6 and 12 months vs. baseline in the placebo group.

Figure 2 shows the percentages of eyes in which the stages of NPDR improved, remained unchanged, or worsened at each study visit compared with baseline. Overall changes in NPDR stages (improvement, no change, worsening) were statistically significant in the placebo group at 6 months (*p* = 0.045) compared with the DHA group, but between-group differences at 12 months (*p* = 0.825), 18 months (*p* = 0.931), and 24 months (*p* = 0.526) did not reach statistical significance.

### 3.3. Secondary Outcome Variables

The results of secondary outcome variables are shown in Table 5. In both study groups, VA almost remained unchanged and significant differences between the DHA and placebo groups along the study were not found. Changes of the serum levels of HbA1c showed significant decreases at the 12- and 18-month visits vs. baseline in patients treated with DHA, and at the 12- and 24-month visits vs. baseline in patients treated with the placebo, although the between-group differences were not statistically significant (Table 5).

In relation to the tolerability of the study supplements, gastrointestinal discomfort was recorded in 13 patients (7.6%), regurgitation in 11 (6.5%), nausea in 5 (2.9%), diarrhea in 2 (1.2%), and vomiting in 1 (0.6%), and other complaints in 13 (7.6%). There were no significant differences in the occurrence of adverse events between the study groups. Based upon the pill count, all participants had taken at least 80% of their capsules.

## 4. Discussion

The present randomized controlled clinical study carried out in patients with NPDR in conditions of daily practice shows that dietary supplementation with high dose DHA compared with a placebo was not associated with statistically significant differences in slowing the progression of any stage of NPDR over a treatment period of 2 years. Overall changes in NPDR stages (improvement, no change, worsening) were statistically significant in the placebo group at 6 months, but between-group differences at 12, 18, and 24 months did not reach statistical significance. Of note, the percentage of eyes with mild NPDR at the end of the study compared with baseline showed a slightly higher increase in the DHA group (from 61.7% to 75.7%) than in the placebo group (from 61.9% to 73.1%). However, the decreases in moderate NPDR stage were considerably greater in the DHA group (from 35.1% to 18.7%) than in the placebo group (from 38.8% to 26%). Among the eyes with a severe NPDR stage, there was an increase of one eye in the DHA group and a decrease of one eye in the placebo group, but the percentage of severe NPDR at baseline was higher in the DHA than in the placebo group (3.2% vs. 1.3%).

Other findings of the study were that VA almost remained unchanged, with slight decreases that may be due to an increase in sclerosis of the crystalline lens during the two years of evolution. Between-group differences in VA were not found at any of the study visits. The mean value of HbA1c of 7–8% indicated an adequate metabolic control in our patients, who had a mean duration of diabetes of 14 years and a high percentage of them presented with associated hypertension and dyslipidemia. We also found that HbA1c values slightly decreased over the study period in both study groups, but differences between the DHA and placebo groups were not observed.

Although it has been shown that increased dietary intake or active supplementation with antioxidants, including *n*-3 PUFAs, has a protective effect on diabetes complications including retinopathy [29,30,31,32,33], there is little information on the potential benefits of antioxidant supplementation, particularly *n*-3 PUFAs in the early stages of DR. The retina is highly susceptive to oxidative stress, due principally to the high content of PUFAs, high oxygen uptake, and glucose oxidation. The set of processes triggered by hyperglycemia, such as a formation of AEG, activation of PKC, and the polyol and hexosamine pathways, provoke oxidative stress, which are also reinforced by oxidative stress in a vicious circle, causing a continuous increase in reactive oxygen species (ROS) and the consequent activation of pathophysiological mechanisms underlying the progression of DR [34,35,36]. However, a few studies have evaluated the effect of supplementation with antioxidants in NPDR, and the results obtained have been inconsistent due to differences in the design and study variables, the characteristics of nutraceutical products, or the duration of supplementation.

The nutraceutical product used in the present study includes a high dose of DHA (1 g), EPA, a mixture of B vitamins, vitamins C and E, lutein, zeaxanthin, and minerals, but none of the previous studies reported in the literature used a nutraceutical supplement of similar composition. In relation to lutein, one of the dietary xanthophyll carotenoids with antioxidant properties, in a randomized double-blind, placebo-controlled trial of 31 patients with NPDR assigned to 10 mg/day of lutein or identical placebo for 36 weeks, a slight improvement in VA was observed in the lutein group [37]. Interestingly, lutein supplementation was shown to improve macular pigment optical density in 100 healthy subjects (200 eyes) from a Mediterranean population, being significantly increased in the presence of DHA, which supports the adjunctive role of DHA for a better lutein availability [38]. In our study, lutein was a component of the nutraceutical compound and was administered at a dose of 9 mg/day, but no apparent effects on visual performance were observed. In a retrospective study of 72 patients with NPDR treated with zeaxanthin for 4 months, the addition of lutein in 36 patients compared to 36 patients who did not receive lutein did not show significant differences in VA, contrast sensitivity, or glare sensitivity [39]. In a study of 97 patients with NPRD followed for 5 years, 56 of whom received an antioxidant supplementation and 41 were included in the placebo group, no changes in BCVA were found [40]. To assess the progression of DR, the authors scored NPDR from 1 (mild), 2 (moderate) to 3 (severe), but the mean differences in the final score compared with baseline were similar in both groups, although in the placebo group statistical significance was reached (supplementation group 2.29 (0.66) vs. 2.53 (0.73), *p* = 0.08; placebo group 2.26 (0.76) vs. 2.65 (0.76), *p* < 0.01) [40]. In the study, the supplementation product did not include *n*-3 PUFAs, and the authors did not evaluate progression according to the characteristic retinography features of the three stages of NPDR, as was done in our study. In another study of 62 patients with mild to moderate NPDR assigned to two matched-age groups, a 6-month treatment with a combination of vitamin E, pycnogenol, and coenzyme Q10 was associated with a reduction of ROS levels [41], but the clinical translation into the reduced progression of NPDR was not evaluated. Finally, in a study of 67 patients randomized to an active multi-component formula containing xanthophyll pigments, antioxidants, and selected botanical extracts (*n* = 39) or placebo (*n* = 28) for 6 months, better visual function (contrast sensitivity, macular pigment optical density, color discrimination, macular threshold perimetry) was reported in the supplemented group [42]. However, the number of patients with mild or moderate NPDR was small, since DR was absent in 37 (55.2%) of the diabetic subjects.

Supplementation with high dose DHA triglyceride was used in none of the aforementioned studies. In a previous experience with the same DHA-based nutraceutical compound given for 90 days to patients with NPDR, improved macular function assessed by microperimetry was found [25]. In the present study, however, it may be argued that although DHA supplementation did not appear to significantly affect the progression of the NPDR stages, improvements in macular function could have been observed since the supplementation product was administered for a prolonged period of time, but measurement of macular function-related variables was not included in the study protocol. On the other hand, total plasma antioxidant capacity (TAC) as an indicator of the antioxidant effect of DHA supplementation was not measured, either. In patients with advanced diabetic retinal disease as DME, intravitreal ranibizumab treatment combined with dietary supplementation with the same high rich DHA triglyceride or placebo was associated with the anatomical improvement of DME (decrease in central subfield macular thickness on OCT) in the DHA group after 3 years of treatment [43,44]. Moreover, differences in plasma TAC and erythrocyte membrane DHA content were statistically significant in favor of the DHA supplementation group. 

Compliance with the nutraceutical supplementation was adequate, and none of the patients discontinued the study because of adverse effects, which occurred in a small percentage of patients. Forty-eight patients (24.7%) discontinued the study, especially between 12 and 24 months after enrolment. The reasons for non-attendance to the study visits were unknown, but significant differences in the baseline data compared to patients who completed the study were not observed.

The present findings should be interpreted considering the limitations of the study, including the single-center design, which may account for the slow rate of recruitment. Moreover, other variables that could reflect the effect of DHA supplementation, particularly improvements in macular function, TAC, or IL-6 levels, were not measured, as the present study was focused on the assessment of progression of any pre-proliferative stage of NPDR in patients with diabetes. At baseline, between-group differences in antidiabetic medication were not found, and although it seems unlikely, a potential influence of diabetes treatment masking the DHA effect cannot be discarded. On the other hand, the higher frequency of type 2 diabetes is explained by the mean age of the population (around 60 years), but baseline differences between the study groups in the percentages of patients with type 1 and type 2 diabetes were not found. It seems unlikely that the type of diabetes may introduce a bias into the study findings.

The strengths of the study are the study design (randomized controlled trial), and the assessment of the long-term effects of a highly concentrated DHA plus xanthophyll carotenoid multivitamin product exclusively in diabetic patients with NPDR, especially given the paucity of studies on antioxidant supplementation in NDPR. The increase in the percentages of patients with mild NPDR at the end of the study was 14% in the DHA group vs. 11.2% in the placebo group, whereas moderate NPDR decreased by 16.4% in the DHA group vs. 10.8% in the placebo group. These differences may indicate a trend towards a greater effect of DHA in slowing the progression of the early stages of NPDR. It may be suggested that the antioxidant and other effects of DHA and other compounds may not be sufficiently selective for targeting the specific underlying pathophysiological mechanisms involved in the incipient stages of NPDR.

## 5. Conclusions

In the present randomized double-blind and placebo-controlled clinical study, the use of a nutraceutical supplement of 1050 g/day of DHA triglyceride, EPA, vitamins, minerals, zeaxanthin, and lutein for 2 years did not appear to influence the slowing of the progression of NPDR. At the end of the study, the increase in eyes with mild NPDR stage and the decrease in eyes with the moderate stage compared with baseline were higher in the DHA group, but differences with the placebo did not reach statistical significance. Among the eyes with a severe NPDR stage, there was an increase of one eye in the DHA group and a decrease of one eye in the placebo group, which is difficult to interpret given the between-group disbalance of severe NPDR stage at baseline. Further studies in patients with NPDR are necessary to clarify the role of antioxidant supplementation in the early stages of DR.

## Figures and Tables

**Figure 1 antioxidants-11-00116-f001:**
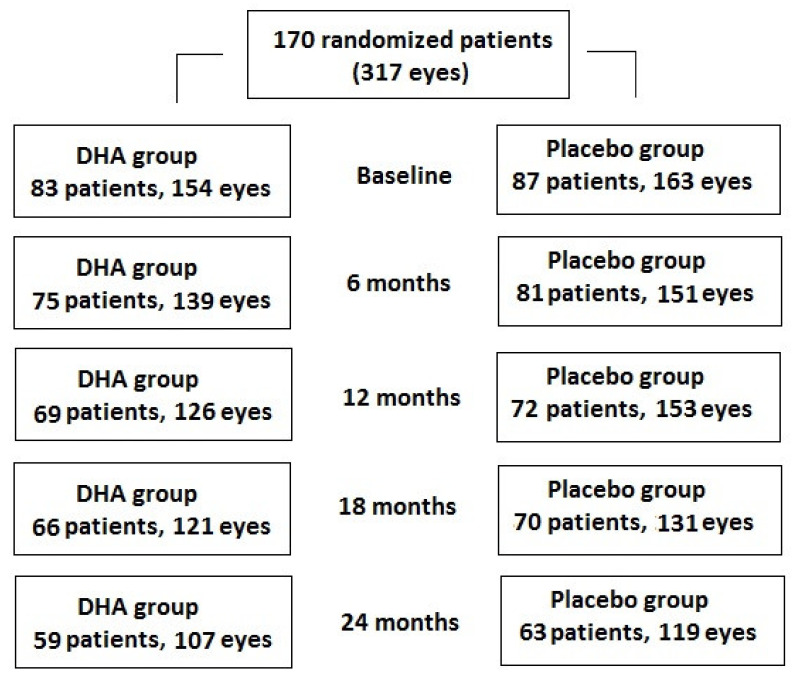
Distribution of patients thought the 2-year study period. Number of patients attending each follow-up visit and number of eyes analyzed.

**Figure 2 antioxidants-11-00116-f002:**
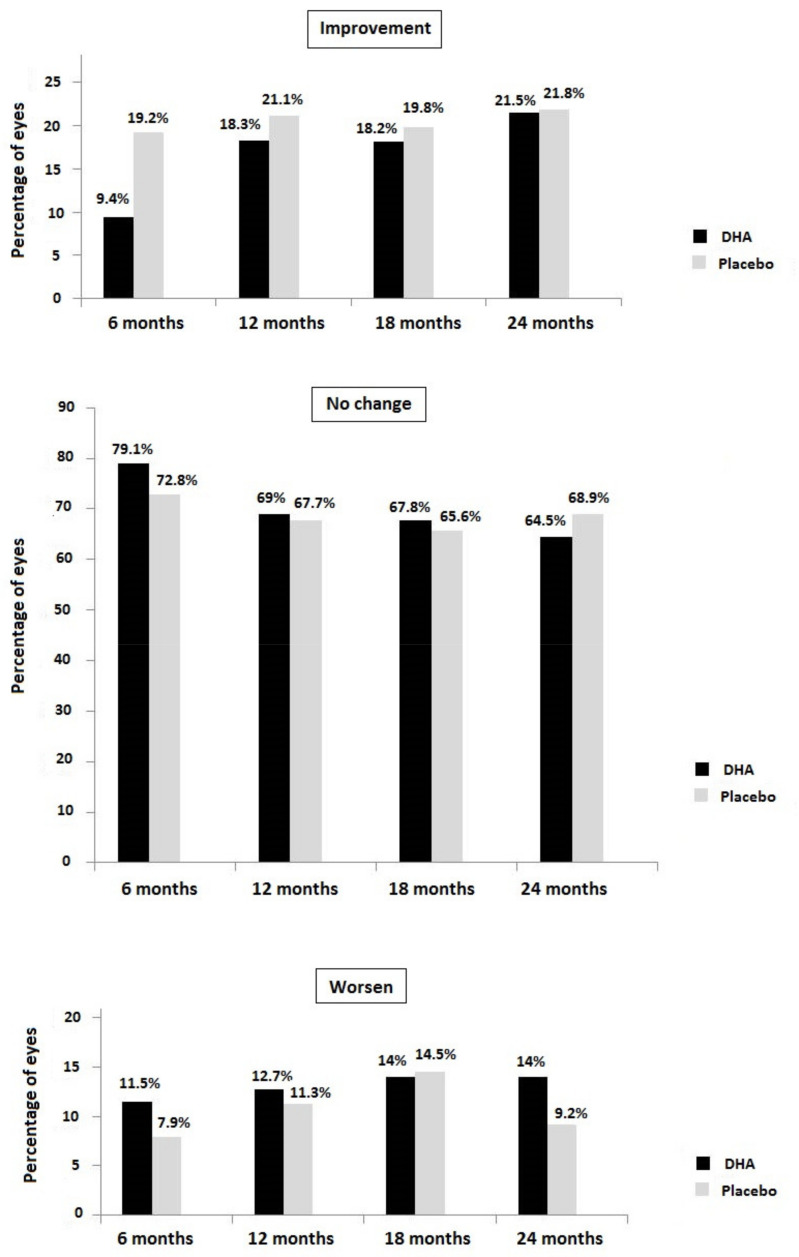
NPDR stage at each study visit in the DHA and placebo groups regarding the percentage of eyes with improvement, no change, or worsening of NPDR stage.

**Table 1 antioxidants-11-00116-t001:** Composition of Brudyretina 1.5 g per capsule.

Composition	Per Capsule	% Recommended Daily Amount	Per Three Capsules	% Recommended Daily Amount
Concentrated oil in ω-3 fatty acids	500 mg		1500 mg	
TG-DHA 70%	350 mg	-	1050 mg	-
EPA 8.5%	42.5 mg	-	127 mg	-
DPA 6%	30 mg	-	90 mg	-
Vitamins				
Vitamin B1 (thiamine)	0.37 mg	33	1.1 mg	100
Vitamin B2 (riboflavin)	0.47 mg	33	1.4 mg	100
Vitamin B3 (niacin/niacinamide)	5.3 mg NE	33	16 mg NE	100
Vitamin B6 (pyridoxine)	0.47 mg	33	1.4 mg	100
Vitamin B9 (folic acid)	66.7 µg	33	200 µg	100
Vitamin B12 (cobalamin)	0.83 µg	33	2.5 µg	100
Vitamin C (ascorbic acid)	26.7 mg	33	80 mg	100
Vitamin E (d-α-TE)	4 mg α-TE	33	12 mg α-TE	100
Essential trace elements				
Zinc	1.66 mg	16.66	5 mg	50
Cooper	0.16 mg	16.66	0.5 mg	50
Selenium	9.16 µg	16.66	27.5 µg	50
Magnesium	0.33 mg	16.66	1 mg	50
Other components				
Lutein	3 mg	-	9 mg	-
Zeaxanthin	0.3 mg	-	0.9 mg	-
Glutathione	2 mg	-	6 mg	-
Energetic value (Kcal)	5.7		17.1	

TG-DHA: triglyceride-bound DHA, DHA: docosahexaenoic acid, EPA: eicosapentaenoic acid, DPA: docosapentaenoic acid, NE: niacin equivalent, TE: tocopherol equivalent.

**Table 2 antioxidants-11-00116-t002:** Demographic and clinical characteristics of the study population.

Variables	Study Group		*p* Value
	DHA (*n* = 83)	Placebo (*n* = 87)	
Gender, males	67 (80.7)	63 (72.4)	0.212
Age, years, mean (SD)	61.7 (10.8)	61.8 (11.8)	0.889
Height, cm, mean (SD)	167.1 (9.0)	168.1 (9.1)	0.393
Weight, kg, mean (SD)	81.9 (12.7)	85.4 (14.8)	0.141
Diabetes			
Type 1	6 (7.2)	9 (10.3)	0.474
Type 2	77 (92.8)	78 (89.7)	
Duration of diabetes, years, mean (SD)	14.9 (9.9)	14.1 (8.4)	0.869
Duration of NPDR, years, mean (SD)	1.0 (1.7)	1.6 (3.2)	0.817
Smoking history			
Current smoker	20 (24.1)	14 (16.1)	0.397
Ex-smoker	34 (41.0)	37 (42.5)	
Never smoker	29 (34.9)	36 (41.4)	
Physical exercise			
Sedentary (none)	22 (26.8)	30 (34.5)	0.512
Moderate (<1 h/day)	24 (29.3)	25 (28.7)	
Active (>1 h/day)	36 (43.9)	32 (36.8)	
Comorbidities			
Hypertension	57 (68.7)	61 (70.1)	0.839
Dyslipidemia	56 (67.5)	49 (56.3)	0.135
Heart disease	8 (9.6)	7 (8.0)	0.714
Nephropathy	1 (1.2)	1 (1.1)	1
Peripheral vascular disease	5 (6.0)	2 (2.3)	0.269
Patients with eyes affected			
Both eyes	72	77	
Left/right	5/6	6/4	
NPDR stage, total eyes	154	163	
Mild	95 (61.7)	101 (61.9)	0.806
Moderate	54 (35.1)	60 (36.8)	0.668
Severe	5 (3.2)	2 (1.3)	0.678
BCVA, mean (SD)			
Left eye			
Decimal	0.748 (0.222)	0.723 (0.261)	0.563
LogMAR	0.160 (0.171)	0.182 (0.172)	0.393
Right eye			
Decimal	0.716 (0.208)	0.752 (0.237)	0.245
LogMAR	0.167 (0.142)	0.161 (0.165)	0.359
HbA1c level, %, mean (SD)	8.38 (1.80)	7.95 (1.68)	0.173
Antidiabetic medication			
Oral antidiabetic agents	43 (51.8)	44 (50.6)	0.293
Insulin	15 (18.1)	9 (10.3)	
Both	25 (30.1)	33 (37.9)	
No medication	0	1 (1.1)	

DHA: docosahexaenoic acid; SD: standard deviation; NPDR: non-proliferative diabetic retinopathy. LogMAR: logarithm of the minimum angle of resolution; BCVA: best-corrected visual acuity.

**Table 3 antioxidants-11-00116-t003:** Demographic and clinical characteristics of patients who completed the study and patients who were lost to follow-up.

Variables	Completed the Study (*n* = 122)	Lost to Follow-Up (*n* = 48)	*p* Value
Gender, males	91 (74.6)	38 (79.2)	0.489
Age, years, mean (SD)	62.4 (11.1)	60.2 (11.8)	0.402
Height, cm, mean (SD)	167.9 (8.7)	166.5 (9.9)	0.339
Weight, kg, mean (SD)	83.5 (14.0)	84.2 (13.7)	0.597
Smoking history			
Current smoker	19 (15.6)	15 (31.2)	0.071
Ex-smoker	54 (44.3)	17 (35.4)	
Never smoker	49 (40.2)	16 (33.3)	
Physical exercise			
Sedentary (none)	39 (32.2)	13 (27.1)	0.439
Moderate (<1 h/day)	37 (30.6)	12 (25.0)	
Active (>1 h/day)	45 (37.2)	23 (47.9)	
Missing	1		
Comorbidities			
Hypertension	80 (65.6)	38 (79.2)	0.083
Dyslipidemia	79 (64.8)	26 (54.2)	0.201
Heart disease	12 (9.8)	3 (6.2)	0.558
Nephropathy	2 (1.6)	0	1
Peripheral vascular disease	6 (4.9)	1 (2.0)	0.675
Patients with eyes affected by NPDR			
Both eyes	104	45	
Left/right	9/9	2/1	
NPDR stage, total eyes	225	92	
Mild	142 (63.1)	54 (58.7)	0.576
Moderate	76 (53.2)	38 (41.3)	0.331
Severe	7 (4.9)	0	
BCVA, mean (SD)			
Left eye			
Decimal	0.756 (0.242)	0.688 (0.238)	0.081
LogMAR	0.163 (0.174)	0.192 (0.165)	0.142
Right eye			
Decimal	0.740 (0.226)	0.718 (0.219)	0.344
LogMAR	0.167 (0.142)	0.170 (0.137)	0.319
HbA1c level, %, mean (SD)	8.08 (1.55)	8.35 (2.18)	0.854
Antidiabetic medication			
Oral antidiabetic agents	57 (46.7)	30 (62.5)	0.293
Insulin	17 (13.9)	7 (14.6)	
Both	47 (38.5)	11 (22.9)	
No medication	1	0	

DHA: docosahexaenoic acid; SD: standard deviation; NPDR: non-proliferative diabetic retinopathy. LogMAR: logarithm of the minimum angle of resolution; BCVA: best-corrected visual acuity.

**Table 4 antioxidants-11-00116-t004:** Changes of NPDR stage in the two study groups.

Study Group and Visits	Total Patients/Eyes	NPDR Stage, Number of Eyes (%)			Within-Group *p* Value *
		Mild	Moderate	Severe	
DHA group					
Baseline	83/154	95 (61.7)	54 (35.1)	5 (3.2)	
6 months	75/139	87 (62.6)	46 (33.1)	6 (4.3)	0.902
12 months	69/126	89 (70.6)	30 (23.8)	7 (5.5)	0.171
18 months	66/126	84 (69.4)	29 (24.0)	8 (6.6)	0.189
24 months	59/121	81 (75.7)	20 (18.7)	6 (5.6)	0.120
Placebo group					
Baseline	87/163	101 (61.9)	60 (36.8)	2 (1.3)	
6 months	81/151	111 (73.5)	40 (26.5)	0	0.025
12 months	72/133	98 (73.7)	29 (21.8)	6 (4.5)	0.004
18 months	70/131	88 (67.2)	43 (32.8)	0	0.275
24 months	63/119	87 (73.1)	31 (26.0)	1 (0.8)	0.084

* McNemar–Bowker test.

**Table 5 antioxidants-11-00116-t005:** Results of secondary outcome variables: visual acuity (VA) and serum levels of HbA1c in the two study groups.

Outcomes	Baseline	6 Months	12 Months	18 Months	24 Months
VA in decimal system					
DHA group, no. eyes	152	139	127	121	107
Mean (SD)	0.73 (0.22)	0.71 (0.24)	0.68 (0.21)	0.64 (0.19)	0.65 (0.22)
Within-group *p* value *		0.147	0.003	<0.001	0.001
Placebo group, no. eyes	160	152	134	129	118
Mean (SD)	0.74 (0.25)	0.71 (0.25)	0.68 (0.26)	0.67 (0.22)	0.69 (0.23)
Within-group *p* value *		0.190	0.003	<0.001	<0.001
Between-group *p* value ^†^	0.732	0.984	0.881	0.372	0.213
VA in logMAR system					
DHA group, no. eyes	152	139	127	120	104
Mean (SD)	0.16 (0.16)	0.19 (0.16)	0.19 (0.14)	0.21 (0.13)	0.21 (0.17)
Within-group *p* value *		0.011	0.021	<0.001	0.012
Placebo group, no. eyes	156	152	134	125	112
Mean (SD)	0.17 (0.17)	0.19 (0.16)	0.21 (0.20)	0.20 (0.14)	0.19 (0.17)
Within-group *p* value *		0.168	0.019	<0.001	0.002
Between-group *p* value ^†^	0.969	0.848	0.689	0.365	0.313
Serum levels of HbA1c, %					
DHA group, no. patients	77	62	60	51	47
Mean (SD)	8.38 (1.80)	7.84 (1.52)	7.69 (1.65)	7.64 (1.12)	7.67 (1.25)
Within-group *p* value *		0.059	0.001	0.010	0.072
Placebo group, no. patients	80	63	67	60	52
Mean (SD)	7.95 (1.68)	7.82 (1.63)	7.68 (1.33)	7.61 (1.25)	7.45 (1.19)
Within-group *p* value *		0.139	0.030	0.124	0.036
Between-group *p* value ^†^	0.173	0.959	0.590	0.627	0.396

* Wilcoxon signed-rank test; ^†^ Mann–Whitney U test; DHA: docosahexaenoic acid; SD: standard deviation. LogMAR: logarithm of the minimum angle of resolution; VA: visual acuity.

## Data Availability

The data presented in this study are available in article.

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
