# Peer review of "Supplementation with a Highly Concentrated Docosahexaenoic Acid (DHA) in Non-Proliferative Diabetic Retinopathy: A 2-Year Randomized Double-Blind Placebo-Controlled Study"

_antioxidants, 2022, doi:10.3390/antiox11010116_

Round 1

Reviewer 1 Report

This study explored the possible utility of supplementing a highly concentrated DHA product with antioxidant activity on non-proliferative diabetic retinopathy in a randomized double-blind placebo controlled study. They conclude that the use of DHA triglyceride nutraceutical supplement for 2 years did not appear to slow the progression of NPDR.

A 2018 article by Gonzalez-Herrero et al on “Supplementation with a highly concentrated docosahexaenoic acid plus xanthophyll carotenoid multivitamin in nonproliferative diabetic retinopathy” found that a high dose DHA and carotenoid treatment in patients significantly improved macular function, total antioxidant capacity and IL-6 levels. Unfortunately none of these significant parameters are included in this long term study thereby dampening the enthusiasm. The parameters included in this study like the changes in the stages of NPDR and HbA1c levels are not statistically significant for the DHA group.

In figure 2, the legend suggests that between-group comparisons for overall changes in NPDR stages at 6 months vs. baseline were statistically significant (P = 0.045). However this data is not discussed clearly in the results and conclusion.

Why the number of eyes analyzed is not close to double the patient number in figure 1? It will be helpful to have the reason for the difference in numbers explained somewhere in the text.

Minor issues:

Abstract section mentions three groups-mild, moderate and severe; but severe group difference in percentage not discussed until the results and conclusion sections. Please include a line about the severe group data in the abstract.

Few sentences were not easy to comprehend. Article needs language editing.  

For example,

  1. abstract line 28-29 is not clear, needs to be rephrased: There were improvements serum levels of HbA1c in both groups but with-out between-group differences.
  2. The distribution of patients thought the study period is shown in Figure 1. A total of -195

Author Response

REVIEWER 1:

This study explored the possible utility of supplementing a highly concentrated DHA product with antioxidant activity on non-proliferative diabetic retinopathy in a randomized double-blind placebo controlled study. They conclude that the use of DHA triglyceride nutraceutical supplement for 2 years did not appear to slow the progression of NPDR.

  • All authors would like to thank the Reviewer for helpful suggestions, which have contributed to improve the quality of the manuscript.

A 2018 article by Gonzalez-Herrero et al on “Supplementation with a highly concentrated docosahexaenoic acid plus xanthophyll carotenoid multivitamin in nonproliferative diabetic retinopathy” found that a high dose DHA and carotenoid treatment in patients significantly improved macular function, total antioxidant capacity and IL-6 levels. Unfortunately none of these significant parameters are included in this long term study thereby dampening the enthusiasm. The parameters included in this study like the changes in the stages of NPDR and HbA1c levels are not statistically significant for the DHA group.

  • Yes, we agree with your comment but in the limitations of the study, we have more clearly recognized this aspect: “Also, other variables that could reflect the effect of DHA supplementation, particularly improvements in macular function, TAC or IL-6 levels were not measured as the present study was focused on assessment of progression of any preproliferative stage of NPDR in patients with diabetes.”

In figure 2, the legend suggests that between-group comparisons for overall changes in NPDR stages at 6 months vs. baseline were statistically significant (P = 0.045). However this data is not discussed clearly in the results and conclusion.

  • To clarify this point, we have rewritten the sentence in the Results section as follows: “Overall changes in NPDR stages (improvement, no change, worsen) were statistically significant in the placebo group at 6 months (P = 0.045) as compared with the DHA group, but between-group differences at 12 months (P = 0.825), 18 months (P = 0.931), and 24 months (P = 0.526) did not reach statistical significance.”
  • In the first paragraph of the Discussion we have added: “Overall changes in NPDR stages (improvement, no change, worsen) were statistically significant in the placebo group at 6 months, but between-group differences at 12, 18, and 24 months did not reach statistical significance.”

Why the number of eyes analyzed is not close to double the patient number in figure 1? It will be helpful to have the reason for the difference in numbers explained somewhere in the text.

  • Not all patients had both eyes affected (Table 1). Figure 1 shows the number of patients (and eyes) who were randomized and those who completed the study and details of eyes corresponding to patients who completed the study and those who were lost to follow-up are shown in Table 2.

Minor issues:

Abstract section mentions three groups-mild, moderate, and severe; but severe group difference in percentage not discussed until the results and conclusion sections. Please include a line about the severe group data in the abstract.

  • We have added this information: “In the DHA group, there were 5 eyes with severe NPDR at baseline, which increased to 1 more at the end of the study. In the placebo group, of 2 eyes with severe NPDR at baseline, 1 eye remained at the end of the study.”

Few sentences were not easy to comprehend. Article needs language editing.  

For example,

  1. abstract line 28-29 is not clear, needs to be rephrased: There were improvements serum levels of HbA1c in both groups but with-out between-group differences.
  • This sentence is rewritten: “There were improvements in serum levels of HbA1c in both groups but significant differences between the DHA and the placebo groups were not found.”

  1. The distribution of patients thought the study period is shown in Figure 1. A total of -195
  • We have added the place of Figure 1 in parenthesis in the text: “A total of 83 patients (154 eyes) were assigned to the DHA group and 59 (71.1%) patients (107 eyes, 69.5%) completed the study, whereas of 87 patients (163 eyes) assigned to the placebo group, 63 (72.4%) (119 eyes, 73.0%) completed the study (Figure 1).”

Reviewer 2 Report

P. Pinas Garcia et al report a prospective randomized double-blind placebo-controlled study (PAOXRED study) spanning two years of DHA supplementation (dose of 1 g/day) in diabetic subjects diagnosed with nonproliferative diabetic retinopathy.

The patients received a nutraceutical product with mostly antioxidant compounds (EPA omega-3, vitamins C and E, zinc, selenium, carotenoids, glutathione) as well as DHA. In addition, they received insulin and diabetes treatment.  This raises the question whether the overall treatment does not mask the DHA effect described in other circumstances cited in the introduction and discussion.

The diabetic patients are mainly type 2. (77-78%).  Could this introduce a bias in the study?

In relation to the pathogenic mechanisms of diabetic retinopathy, how can we interpret the opposite effects of a middle (12-14% increase) and a moderate (10-15% decrease) NPDR?

This DHA supplementation protocol and patient outcomes are difficult to compare to other studies cited in the discussion and therefore interpretation of the results remains unclear.

The originality and strength of this study is the different stages of the disease studied. What is surprising is the lack of effect of DHA on the orientation and progression of the disease. To the point sthat the placebo has more positive or negative effect depending on the stage than DHA.

Author Response

REVIEWER 2

  1. Pinas Garcia et al report a prospective randomized double-blind placebo-controlled study (PAOXRED study) spanning two years of DHA supplementation (dose of 1 g/day) in diabetic subjects diagnosed with nonproliferative diabetic retinopathy.
  • We would like to thank you for time and valuable comments regarding the scientific interest of the study.

The patients received a nutraceutical product with mostly antioxidant compounds (EPA omega-3, vitamins C and E, zinc, selenium, carotenoids, glutathione) as well as DHA. In addition, they received insulin and diabetes treatment.  This raises the question whether the overall treatment does not mask the DHA effect described in other circumstances cited in the introduction and discussion.

  • The possibility of the effect of antidiabetic medication on the results obtained cannot be excluded, although it seems unlikely as differences in antidiabetic medication at baseline were not significantly different between the study groups. However, we have added a comment in the Discussion: “At baseline, between-group differences in antidiabetic medication were not found and although it seems unlikely, a potential influence of diabetes treatment masking the DHA effect cannot be discarded.”

The diabetic patients are mainly type 2. (77-78%).  Could this introduce a bias in the study?

  • “On the other hand, the higher frequency of type 2 diabetes is explained by the mean age of the population (around 60 years), but baseline differences between the study groups in the percentages of patients with type 1 and type 2 diabetes were not found. It seems unlikely that the type of diabetes may introduce a bias in the study findings.” We have added this comment in the Discussion.

In relation to the pathogenic mechanisms of diabetic retinopathy, how can we interpret the opposite effects of a middle (12-14% increase) and a moderate (10-15% decrease) NPDR?

  • We have added this comment in the Discussion: “The increase in the percentages of patients with mild NPDR at the end of the study was 14% in the DHA group vs. 11.2% in the placebo group, whereas moderate NPDR decreased by 16.4% in the DHA group vs.10.8% in the placebo group. These differences may indicate a trend towards a greater effect of DHA in slowing the progression of early stages of NPDR. It may be suggested that the antioxidant and other effects of DHA and other compounds may not be sufficiently selective for targeting the specific underlying pathophysiological mechanisms involved in the incipient stages of NPDR.”

This DHA supplementation protocol and patient outcomes are difficult to compare to other studies cited in the discussion and therefore interpretation of the results remains unclear.

  • Interpretation of results in the light of data reported in previous publications is difficult since to our knowledge this is the first study focused on the effect of DHA supplementation on NPDR. This is stated at the end of the Introduction and justifies the study: “To our knowledge, clinical studies on the effect of DHA dietary supplementation with antioxidant activity in diabetic patients with NPDR have not been previously reported. Therefore, a randomized controlled study was conducted to determine whether dietary supplementation with high dose DHA for 2 years could slow the progression of any pre-proliferative stage of NPDR in patients with diabetes.”

The originality and strength of this study is the different stages of the disease studied. What is surprising is the lack of effect of DHA on the orientation and progression of the disease. To the point that the placebo has more positive or negative effect depending on the stage than DHA.

As we have commented on previously “The increase in the percentages of patients with mild NPDR at the end of the study was 14% in the DHA group vs. 11.2% in the placebo group, whereas moderate NPDR decreased by 16.4% in the DHA group vs.10.8% in the placebo group. These differences may indicate a trend towards a greater effect of DHA in slowing the progression of early stages of NPDR.”

Reviewer 3 Report

Dear Authors, 

this paper is worthy to publish, apart from the negative results. The paper is well designed, well prepared, the study design and results are well described. I have some minor points to improve the legibility. 

Abstract:
- please change "the effect of 2-year supplementation" for "the effect of a 2-year supplementation"  (p.1, l. 18)
- DHA - explain the abbreviation (p. 1, l. 18)
- add a comma before and "mild, moderate and severe stages" (p.1, l. 22)
- change "There were improvements serum levels"  for "in serum" (p.1, l.28)
Introduction:
- change "About one third" for "About one-third" (p.1, l. 44)
- change "any preproliferative stage" for "any pre proliferative stage" (p.2, l. 88)

Methods:
- delate a bracket in the phrase "appointment at the study center. )" (p. 2, l.97)
- add a comma before and in the phrase "mild, moderate and severe stages" (p.3, l.98)
- change "50% probability" for "a 50% probability" (p.3, l. 118)
- add a comma before and in the phrase "at 6, 12 and 18 months" (p.4, l. 152) 

Results:
- page 6, line 193 - please explain the abbreviations: LogMAR, BCVA
- please consider the change the place of Figure 1 immediately after the sentence "The distribution of patients thought the study period is shown in Figure 1." (p.6, l. 195)
- page 6, line 206 - please explain the abbreviations: LogMAR, BCVA
- change "comparison of visit at 6 and 12 months" for "comparison of visits at 6 and 12 months" (p.7, l. 218)
- page 10, line 240 - please explain the abbreviations: LogMAR, VA
- change the phrase "In relation to tolerability of the study supplements" to "In relation to the tolerability of the study supplements" (p.10, l.241)

Discussion:
- change the phrase "such as formation of AEG" for "such as a formation of AEG" (p.11, l. 273)
- change "translation into reduced progression" for "translation into the reduced progression" (p. 11, l, 307) 
- change "with anatomical improvement of DME" for "with the anatomical improvement of DME" (p.12, l.326)

In the present randomized double-blind and placebo-controlled clinical study, the 347
use of a nutraceutical supplement of 1,050 g/day of DHA triglyceride, EPA, vitamins, min- 348
erals, and zeaxanthin and lutein for 2 years did not appear to influence on slowing the 349
progression of NPDR. At the end of the study, the increase in eyes with mild NPDR stage 350
and the decrease in eyes 

- change "with moderate stage" for "with the moderate stage" (p. 12, l. 351)
 as compared with baseline was higher in 351
the DHA group, but differences with placebo did not reach

- change "statistical significant" for "statistical significance" (p.12, l. 352)

Thank you. 

Author Response

REVIEWER 3

Dear Authors, 

this paper is worthy to publish, apart from the negative results. The paper is well designed, well prepared, the study design and results are well described. I have some minor points to improve the legibility. 

  • Thank you very much, we appreciate all this minor corrections to improve legibility.

Abstract:
- please change "the effect of 2-year supplementation" for "the effect of a 2-year supplementation"  (p.1, l. 18)
. Corrected.
- DHA - explain the abbreviation (p. 1, l. 18). Corrected.
- add a comma before and "mild, moderate and severe stages" (p.1, l. 22)
. Comma added.
- change "There were improvements serum levels"  for "in serum" (p.1, l.28)
. Corrected.

Introduction:
- change "About one third" for "About one-third" (p.1, l. 44)
. Corrected.
- change "any preproliferative stage" for "any pre proliferative stage" (p.2, l. 88)
. Corrected.

Methods:
- delate a bracket in the phrase "appointment at the study center. )" (p. 2, l.97).
Deleted.
- add a comma before and in the phrase "mild, moderate and severe stages" (p.3, l.98)
. Comma added.
- change "50% probability" for "a 50% probability" (p.3, l. 118)
. Corrected.
- add a comma before and in the phrase "at 6, 12 and 18 months" (p.4, l. 152)
. Comma added. 

Results:
- page 6, line 193 - please explain the abbreviations: LogMAR, BCVA
. Abbreviations are explained.

- please consider the change the place of Figure 1 immediately after the sentence "The distribution of patients thought the study period is shown in Figure 1." (p.6, l. 195). In response to comments of another Reviewer, we have included Figue 1 in parenthesis in this sentence: “A total of 83 patients (154 eyes) were assigned to the DHA group and 59 (71.1%) patients (107 eyes, 69.5%) completed the study, whereas of 87 patients (163 eyes) assigned to the placebo group, 63 (72.4%) (119 eyes, 73.0%) completed the study (Figure 1).”

- page 6, line 206 - please explain the abbreviations: LogMAR, BCVA
. Abbreviations are explained.
- change "comparison of visit at 6 and 12 months" for "comparison of visits at 6 and 12 months" (p.7, l. 218)
. Visits, OK.
- page 10, line 240 - please explain the abbreviations: LogMAR, VA
. Abbreviations are explained.
- change the phrase "In relation to tolerability of the study supplements" to "In relation to the tolerability of the study supplements" (p.10, l.241)
. Corrected.

Discussion:
- change the phrase "such as formation of AEG" for "such as a formation of AEG" (p.11, l. 273)
. Corrected.
- change "translation into reduced progression" for "translation into the reduced progression" (p. 11, l, 307)
. Corrected. 
- change "with anatomical improvement of DME" for "with the anatomical improvement of DME" (p.12, l.326)
. Corrected.

In the present randomized double-blind and placebo-controlled clinical study, the 347
use of a nutraceutical supplement of 1,050 g/day of DHA triglyceride, EPA, vitamins, min- 348
erals, and zeaxanthin and lutein for 2 years did not appear to influence on slowing the 349
progression of NPDR. At the end of the study, the increase in eyes with mild NPDR stage 350
and the decrease in eyes 

- change "with moderate stage" for "with the moderate stage" (p. 12, l. 351)
 as compared with baseline was higher in 351
. Corrected.
the DHA group, but differences with placebo did not reach- change "statistical significant" for "statistical significance" (p.12, l. 352)
. Corrected.

Thank you. 

Round 2

Reviewer 1 Report

Concerns have been addressed.

Reviewer 2 Report

The manuscript has been sufficiently improved by following the reviewer's recommendations.